# The Relationship between *Brachionus calyciflorus*-Associated Bacterial and Bacterioplankton Communities in a Subtropical Freshwater Lake

**DOI:** 10.3390/ani12223201

**Published:** 2022-11-18

**Authors:** Yongzhi Zhang, Sen Feng, Fan Gao, Hao Wen, Lingyun Zhu, Meng Li, Yilong Xi, Xianling Xiang

**Affiliations:** 1School of Ecology and Environment, Anhui Normal University, Wuhu 241002, China; 2Collaborative Innovation Center of Recovery and Reconstruction of Degraded Ecosystem in Wanjiang Basin Co-Founded by Anhui Province and Ministry of Education, Wuhu 241002, China

**Keywords:** *Brachionus calyciflorus*, associated bacteria, bacterioplankton, diversity, functional traits

## Abstract

**Simple Summary:**

This study explored the relationship between *Brachionus calyciflorus*-associated bacterial and bacterioplankton communities in freshwater. We believe that our study makes a significant contribution because zooplankton and bacterioplankton are the basic components in the aquatic ecosystem, zooplankton has an important role between lower (phyto-, protozooplankton) and higher (fish) trophic levels, and bacteria participate in biogeochemical cycle processes such as nitrogen and carbon cycle, where the symbiotic relationship between them plays an important role in the nutrient cycle, so researching the symbiotic relationship between them will contribute to monitoring the process of environmental change and ecological restoration. Overall, our study expands the current understanding of zooplankton–bacteria interaction and promotes the combination of two different research fields.

**Abstract:**

Zooplankton bodies are organic-rich micro-environments that support fast bacterial growth. Therefore, the abundance of zooplankton-associated bacteria is much higher than that of free-living bacteria, which has profound effects on the nutrient cycling of freshwater ecosystems. However, a detailed analysis of associated bacteria is still less known, especially the relationship between those bacteria and bacterioplankton. In this study, we analyzed the relationships between *Brachionus calyciflorus*-associated bacterial and bacterioplankton communities in freshwater using high-throughput sequencing. The results indicated that there were significant differences between the two bacterial communities, with only 29.47% sharing OTUs. The alpha diversity of the bacterioplankton community was significantly higher than that of *B. calyciflorus*-associated bacteria. PCoA analysis showed that the bacterioplankton community gathered deeply, while the *B. calyciflorus*-associated bacterial community was far away from the whole bacterioplankton community, and the distribution was relatively discrete. CCA analysis suggested that many environmental factors (T, DO, pH, TP, PO_4_^3-^, NH_4_^+^, and NO_3_^-^) regulated the community composition of *B. calyciflorus*-associated bacteria, but the explanatory degree of variability was only 37.80%. High-throughput sequencing revealed that *Raoultella* and *Delftia* in Proteobacteria were the dominant genus in the *B. calyciflorus*-associated bacterial community, and closely related to the biodegradation function. Moreover, several abundant bacterial members participating in carbon and nitrogen cycles were found in the associated bacterial community by network analysis. Predictive results from FAPROTAX showed that the predominant biogeochemical cycle functions of the *B. calyciflorus*-associated bacterial community were plastic degradation, chemoheterotrophy, and aerobic chemoheterotrophy. Overall, our study expands the current understanding of zooplankton–bacteria interaction and promotes the combination of two different research fields.

## 1. Introduction

In a freshwater ecosystem, zooplankton and bacterioplankton are fundamental components of the food web. Zooplankton plays an important role between lower (phyto-, protozooplankton) and higher (fish) trophic levels [1]. Bacteria participate in biogeochemical cycle processes such as ammonia oxidation [2], nitrogen cycle [3], and carbon cycle [4]. Generally, they are viewed as independent study objects that are only indirectly connected via nutrient cycling and trophic cascades [5]. Until recently, attention has been paid to bacteria that colonize inside and outside zooplankton, known as zooplankton-associated bacteria [6,7].

Some studies have pointed out that the community structure of zooplankton-associated bacteria is mainly determined by the species-specific characteristics of the host and the environmental characteristics of the host (including food and the surrounding bacterial community) and has a similar community composition but a different relative abundance than the bacterioplankton in the surrounding water [8,9]. Bacterioplankton can use organic matter released by zooplankton as a carbon and nitrogen source for survival and reproduction [10]. Zooplankton provides a unique microhabitat by releasing bioavailable dissolved organic matter and nutrients during digestion [6,11]. Driven by different physical and chemical conditions, a unique associated bacterial community was constructed [12]. The composition of the zooplankton-associated bacterial community depends on the bacterial pool in the surrounding water, and zooplankton migrate vertically in layered water, which is convenient for associated bacteria to obtain and spread from different water layers [13].

It has been reported that changes in environmental factors, such as temperature and nutrients, will affect the composition and diversity of the bacterioplankton community [14], and thus affect those in the zooplankton-associated bacterial community [9]. Conversely, different associated bacterial community compositions will also have different effects on the population dynamics of the host. Callens and others [15] used antibiotics to interfere with the microbial community in the culture environment and found that they had a strong impact on the subsequent colonization of *Daphnia magna*-associated bacteria, and then affected the growth of *D. magna*. These studies indicate that the composition of bacterioplankton in environmental water may be an important factor in the construction of the zooplankton-associated bacterial community, which has an important impact on the growth and reproduction of hosts. There is an active bacterial exchange process between zooplankton-associated bacteria and bacterioplankton, and the close relationship between them can widely affect the behavior, growth, and biogeochemical activities of bacteria [16]. Therefore, the exploration of the relationship between bacterioplankton and associated bacterial communities is helpful to better understand how native bacterial communities shape the composition of associated bacteria and affect the interaction between host and bacterial communities, and also has important scientific significance for understanding the adjustment of the whole aquatic ecosystem function.

Although the close relationship between copepod, cladoceran, and bacteria has been widely studied [12,17,18], studies on rotifer-associated bacteria remain scarce. Rotifers can account for between 10% and 44% of total zooplankton production [19], and thus play an important role as herbivores suspension breeders and predators in zooplankton communities. Additionally, some scholars believe that the habitat’s bacterial community composition (BCC) influences rotifer culture stability and population growth in the laboratory [20,21]. These studies only analyzed the bacterial community in water, while the analysis of rotifer-associated bacteria was little or not detailed, especially the analysis of the relationship between rotifer-associated bacteria and bacterioplankton in freshwater was even less.

As a cosmopolitan rotifer species, *B. calyciflorus* was the subject of this study and is often used as a model organism. Most studies have focused on food restriction [22], toxicity tests [23], and interspecific competition [24]. However, few studies have focused on the relationship between *B. calyciflorus*-associated bacteria and bacterioplankton communities. Although the bacterioplankton community serves as a source library for zooplankton-associated bacterial communities, given the high sensitivity of the bacterial community structure to environmental changes and the fact that *B. calyciflorus* provides a microhabitat distinct from the surrounding water, we hypothesized that the *B. calyciflorus*-associated bacterial community structure is significantly different from the bacterioplankton community structure. Furthermore, we hypothesized that the physical and chemical factors in the surrounding water were not the main factors regulating the *B. calyciflorus*-associated bacterial community structure. To test these hypotheses, our study used high-throughput sequencing technology to clarify taxonomic and functional information of *B. calyciflorus*-associated bacterial and bacterioplankton communities and analyzed them in combination with physicochemical factors in the surrounding water.

## 2. Materials and Methods

### 2.1. Sample Collection and Treatment

All samples were collected from Lake Jinghu (31°19′45″ N, 118°22′29″ E) in Wuhu city, Anhui, China, where the average depth of water is 2 m, and all field sampling work was conducted from June 2021 to May 2022. We identified *B. calyciflorus* by taxonomic macroscopic characteristics [25]. When sampling work was carried out every month, water temperature (T), dissolved oxygen (DO), and pH were measured by a YSI multi-parameter water quality analyzer (YSI 6600, Yellow Springs, OH, USA). Mixed layers of water samples were collected to perform physicochemical and water bacterial community structure analysis. Determination of TN, TP, PO_4_^3−^, NH_4_^+^, and NO_3_^−^ was completed by the Taihu Lake Ecosystem Experimental Station of the National Academy of Sciences. Total nitrogen (TN) was determined by the alkaline potassium persulfate digestion UV spectrophotometric method, ammonia nitrogen (NH_4_^+^) by Nessler’s reagent spectrophotometry, nitrate nitrogen (NO_3_^−^) by UV spectrometry, and total phosphorus (TP) and phosphate (PO_4_^3-^) by the ammonium molybdate spectrophotometric method [26].

In addition, the maximum number of *B. calyciflorus* were sampled to drag horizontally and vertically with a plankton net (64 µm mesh), and a total of 7.5 L of well-mixed water samples were collected from 0.5 m, 1 m, and 1.5 m in the lake using a glass water collector (size 2.5 L), respectively. Subsequently, all samples were immediately taken back to the laboratory to obtain *B. calyciflorus*-associated bacterial (AB) and bacterioplankton (NB) samples, thereby avoiding great changes in the microbiota. Undamaged, actively swimming *B. calyciflorus* were picked out, rinsed with sterilized water 3~5 times to remove loosely attached bacteria, then used for isolation of total DNA, and finally transferred into 1.5 mL sterile centrifuge tubes. In particular, due to the small size of *B. calyciflorus* and the limited biomass of the associated bacterial community, to ensure a sufficient sample size for bacterial sequencing, it is necessary to pick more than 500 similar individuals in each sample. In addition, *B. calyciflorus*-associated bacteria include all bacteria attached to the outer surface and resident in the gut, so bacterial DNA was extracted from the entire individual of *B. calyciflorus*. At the same time, 500 mL mixed water samples were pumped through a 0.22 μm membrane (Shanghai Sangon Biotech, F513134), then used for isolation of total DNA, and placed into 1.5 mL sterile centrifuge tubes. All obtained bacterial samples were preserved below −80 °C until further analysis.

### 2.2. Sequence Determination of the 16S Gene from the Bacterial Community

Genomic DNA of bacterial communities was extracted from samples using the E.Z.N.A.^®^ Soil DNA Kit (Omega Biotek, Norcross, GA, USA). The V3-V4 hypervariable region of the bacterial 16S rRNA gene was amplified by PCR thermal cycling apparatus (GeneAmp^®^ 9700, ABI, USA) using primers 338F (5′-ACTCCTACGGGAGGCAGCAG-3′) and 806R (5′-GGACTACHVGGGTWTC-TAAT-3′). PCR amplification was carried out using TransStart Fastpfu DNA Polymerase (TransGen AP221-02) in a 20 μL reaction system, and amplification parameters were as follows: the initial denaturation at 95 °C lasted for 3 min, then denaturation at 95 °C for 30 s, annealing at 55 °C for 30 s, and extension at 72 °C for 45 s, 27 cycles in total, with a single extension at 72 °C for 10 min and termination at 10 °C. The PCR product was extracted from a 2% agarose gel, purified using the AxyPrep DNA Gel Extraction Kit (AxyPrep Biosciences, Union City, NJ, USA), and quantified using a Quantus™ fluorometer (Promega, Madison, WI, USA). According to the standard protocol of Majorbio BioPharm Technology Co., (Shanghai, China), amplicons were collected and purified in an equimolar and double-terminal manner on the Illumina MiSeq PE300 platform (Illumina, San Diego, CA, USA). UPARSE (version 7.1 http://www.drive5.com/-uparse/ accessed on 20 June 2022) was used to cluster operational taxons (OTUs) with a 97% similarity cut-off value, and chimeric sequences were identified and deleted. The classification of each OTU representative sequence after removing singletons was analyzed against the Silva database (version 138 https://www.ar-bsilva.de/ accessed on 20 June 2022) using a 70% confidence threshold using the RDP classifier (version 2.11 https://sourceforge.net/projects/rdpclassifier/ accessed on 20 June 2022).

### 2.3. Data Analysis

The data of high-throughput sequencing analysis and mapping were carried out with the help of the Majorbio Online Cloud Platform (http://www.majorbio.com/ accessed on 20 June 2022). Using Qiime (version 1.9.1 http://qiime.org/install/index.html accessed on 20 June 2022) to calculate alpha diversity index under different random sampling, using R language (version 3.3.1) and Python to complete sparse curve analysis, PCoA analysis, Venn diagram, species composition analysis, heatmap, intergroup difference test, co-occurrence network and FAPROTAX (FAPROTAX v1.2.1, http://www.loucalab.com/archive/FAPROTAX/lib/php/index.php?section=Download accessed on 20 June 2022) function prediction analysis and mapping.

The other data were analyzed using SPSS 22.0 and expressed as mean ± SD (standard deviation). The one-sample Kolmogorov–Smirnov procedure and Levene’s test were used to test the data for normality and homogeneity of variances, respectively. For the parameters with significant effects, multiple comparisons were conducted to identify which groups were significantly different among the treatments. For data with homogenous variance, Duncan’s method was selected to compare the differences among groups, and the data with uneven variance were compared using the Games–Howell method. In the process of variance analysis, the data that did not conform to the normal distribution were adjusted logarithmically. The formula for data logarithmic processing was as follows [27]:*Y_i_* = *ln*(*x_i_* + 1)(1)
where *x_i_* is the original value and *Y_i_* is the converted value.

## 3. Results

### 3.1. Alpha and Beta Diversity of AB and NB Communities

A total of 3,518,091 bacterial sequences were obtained from 24 samples involving 12 months, with an average read length of 416 and a total number of 4621 OTUs. The sequencing results showed that the effective sequence number of some samples of *B. calyciflorus*-associated bacteria in June, July, September, and November was too low, so it is necessary to reject abnormal data. Meanwhile, the chloroplasts, mitochondria, and species with a total sequence number of less than 10 need to be removed, and then the minimum sample sequence number of 6303 was selected for subsampling analysis to finally obtain the total number of 43 phylum, 121 class, 184 order, 457 family, 840 genus, and 2428 OTUs.

Sparse analysis showed that the sparse curve of the AB samples was close to saturation, while the NB samples did not tend to be flat (Appendix A), indicating that the AB samples were enough to complete the sequencing of most members of the bacterial community, while the NB samples were insufficient. The sequencing coverage of AB samples reached 99%, indicating that the sequencing depth was enough to cover most bacteria, including rare species, while the sequencing coverage of NB samples only reached 97% (Table 1). The phylogenetic diversity index (PD), species richness (Chao), and Shannon diversity index of the AB group were significantly lower than the NB group (*p* < 0.001) (Table 1). These results indicated that the diversity of the AB community was significantly lower than the NB community (*p* < 0.001). In addition, alpha diversity was not significantly different between summer (June to August) and autumn (September to November) in both AB and NB groups, and the Shannon diversity index was significantly lower in winter (December to February) than in both summer and autumn (*p* < 0.05, Appendix A).

PCoA based on Bray-Curtis distance was used to study the community difference of the AB and NB samples. PCoA clearly distinguishes between AB and NB communities (Figure 1). According to the Bray–Curtis dissimilarity, the *B. calyciflorus*-associated bacterial (AB) communities were taxonomically distant from the bacterioplankton (NB) communities. The difference is that the NB community is more tightly clustered (Figure 1).

### 3.2. Differences in Community Composition of the AB and NB Groups

The AB community was mainly composed of Proteobacteria (76.67 ± 12.67%), Actinobacteriota (11.83 ± 7.27%), Cyanobacteria (4.05 ± 2.37%), Verrucomicrobiota (2.16 ± 1.48%), Firmicutes (1.44 ± 1.30%) and Bacteroidota (1.44 ± 0.69%) (Figure 2A). In the composition of the NB community, Actinobacteriota (33.94 ± 10.42%) had the highest relative abundance, followed by Cyanobacteria (22.38 ± 9.52%), Proteobacteria (18.08 ± 3.91%), Bacteroidota (8.24 ± 2.51%), Firmicutes (5.70 ±1 1.64%), Verrucomicrobiota (4.89 ± 1.83%), and so on (Figure 2A). Proteobacteria not only dominated the AB community but also the only taxa whose relative abundance was higher than that of the NB community (Figure 2A). Proteobacteria were mainly composed of Enterobacterales, Burkholderiales, Pseudomonadales, Rickettsiales, and Rhizobiales (Figure 2B). Actinobacteriota was the most important bacterial taxa of bacterioplankton, which was mainly composed of Microtrichales, and Frankiales (Figure 3). The relative abundance of Cyanobacteria was second among the AB and NB communities, which were mainly composed of Synechococcales (Figure 2B). Furthermore, in both groups AB and NB the relative abundance of Proteobacteria was higher in spring and winter than in summer and autumn, while Actinobacteriota and Cyanobacteria were lower in spring and winter than in summer and autumn (Appendix A).

The community composition of the AB and NB groups was quite different, and Proteobacteria was the most significant taxa (Figure 2). After analyzing the differences of the top 15 genera (Student’s *t*-test) by the difference significance test between groups, it was found that there were significant differences in 13 genera (*p* < 0.05). Among the 13 genera with significant differences, only the relative abundance of *Raoultella* and *Delftia* was significantly higher in the AB group than in the NB group (Figure 3). The relative abundance of *Pseudomonas* and *Candidatus_Megaira* was also higher in the AB group.

### 3.3. Significantly Different OTUs between AB and NB Communities

The number of OTUs shared by the AB and NB communities was 974 (40.12%), while the number of unique OTUs was 418 (17.22%) and 1036 (42.67%), respectively (Figure 4A). At the same time, the NB community has more abundant OTUs, which further illustrates its higher alpha diversity. The shared OTUs were mainly composed of *Raoultella* (OTU2508), *Delftia* (OTU3989), *Cyanobium* PCC-6307 (OTU2205, 2255, and 4176), and the *hgcI clade* (OTU2852 and 2456) (Figure 4B). However, among shared OTUs, the distribution of species abundance in the AB and NB communities was not matched. For example, OTU2508 and OTU3989 were the only OTU in *Raoultella* and *Delftia*, respectively, and their relative abundance was the highest in the AB community but lower in the NB community (Figure 4). Among the unique OTUs of the AB community, *Candidatus Hepatincola* (OTU118) and *Tyzzerella* (OTU1258) were abundant (Figure 4B). Among the unique OTUs of the NB community, there was little difference between OTUs with relative abundance above 1% (Figure 4B). Surprisingly, the OTUs with a relative abundance of less than 1% were 54.66% in the unique OTUs of the AB community and as high as 81.64% in the NB community (Figure 4B).

### 3.4. Relationship between Bacterial Community and Environmental Parameters

There are many environmental factors related to community distribution, but many of them have strong collinearity. Variance Inflation Factor (VIF) analysis was a commonly used method for screening environmental factors and computed as VIF_j_ = 1/(1 − R_j_^2^), R_j_^2^ represents the proportion of variance of the jth independent variable related to other independent variables in the model [28]. In general, a VIF greater than 10 indicates that the regression model has severe multicollinearity. Through VIF variance expansion factor analysis, the environmental factors with VIF greater than 10 were screened and removed, and multiple screenings were performed until the VIF values corresponding to the selected environmental factors were all less than 10. Finally, the environmental factors associated with AB and NB communities were screened out as T, DO, pH, TN, TP, PO_4_^3−^, NH_4_^+^, and NO_3_^-^ (Appendix A). After the DCA analysis, the CCA model was selected to analyze the correlation between bacterial communities and environmental factors. As shown in Figure 5, the two bacterial communities had different responses to environmental factors. The T, DO, pH, TP, PO_4_^3−^, NH_4_^+^, and NO_3_^−^ were significantly correlated with the variation of AB community, and all the selected parameters on the first two axes explained 39.55% of the bacterial community change (Figure 5A). The T, DO, TP, PO_4_^3−^ and NO_3_^−^ were significantly correlated with the variation of the NB community, and all the selected parameters on the first two axes explained 39.69% of the bacterial community change (Figure 5B).

### 3.5. Co-Occurrence Network of AB and NB Communities

To infer the potential keystone taxa of AB and NB communities at OTUs, a co-occurrence network was calculated based on Spearman r (correlation function, *p* < 0.05), and the correlations with absolute r values above 0.6 were retained (Figure 6). The top 50 dominant taxa of relative abundance at the order level were selected for the network analysis. We selected the values that can screen out the top 10 of the highest degrees, closeness centrality, and betweenness centrality as thresholds for defining keystone taxa in bacterial communities. For the AB group, OTUs with degree centrality, closeness centrality, and betweenness centrality higher than 0.55, 0.62, and 0.03, respectively, were selected as the keystone taxa. For the NB group, OTUs with degree centrality, closeness centrality, and betweenness centrality higher than 0.51, 0.51, and 0.05, respectively, were selected as the keystone taxa. Three keystone taxa were speculated from both network structures. The keystone taxa of the AB community were composed of Frankiales (OTU2852), Microtrichales (OTU2544), and Gammaproteobacteria_Incertae_Sedis (2845). The keystone taxa of the NB community were composed of Frankiales (OTU2852 and 2823), Microtrichales (OTU4176), and Burkholderiales (OTU2398) (Figure 6 and Appendix A). We counted the edges in the network and found that the AB community had 336 edges, of which 268 were positive and 68 were negative correlations, while the NB community had 256 edges, of which 160 were positive and 96 were negative correlations (Appendix A).

### 3.6. Functional Characteristics of AB and NB Communities

Many microorganisms are involved in key biogeochemical processes and interspecific interactions. FAPROTAX function prediction is mainly used to further analyze the biogeochemical cycle function of microorganisms, especially the cycle function of sulfur, carbon, hydrogen, and nitrogen. Through FAPROTAX function prediction analysis (based on the Wilcoxon rank-sum test), 37 biogeochemical cycle functions were predicted from the AB and NB communities (Figure 7A). Although phototrophy, photoautotrophy, oxygenic photoautotrophy, and plastic degradation were the most common functions among the AB and NB communities, there were significant differences in functional abundance between the two bacterial communities (Figure 7). Among the most mainly biogeochemical functions of the AB community were plastic degradation, chemoheterotrophy, and aerobic chemoheterotrophy, while autotrophic functions, such as phototrophy, photoautotrophy, and oxygen photoautotrophy, were more prominent in the NB community. In addition, animal parasites or symbionts, human pathogens all, and human pathogen pneumonia were abundant in the NB community (Figure 7B).

## 4. Discussion

### 4.1. Community Composition Difference and Keystone Taxa between AB and NB Communities

In this study, the communities of AB and NB were compared to explore the relationship between the two bacterial communities. Although they differed significantly from each other, there was still 40.12% shared OTU between them, indicating that bacterial exchange occurred between *B. calyciflorus* and environmental water. Some studies have pointed out that zooplankton-associated bacteria community structures are determined by the species-specific characteristics and contacted environmental characteristics (including food and surrounding bacterial community) of the host [9]. Although the *B. calyciflorus*-associated bacterial community contains abundant rare OTU, which was not detected in environmental water, this might be caused by insufficient coverage of the detection of the bacterioplankton community. On the contrary, many bacteria in environmental water fail to colonize in *B. calyciflorus* bodies, probably because their survival requirements do not match the conditions and resources provided by the host [29].

Proteobacteria was the most important taxon in the *B. calyciflorus*-associated bacterial community. Studies have shown that members of Proteobacteria are ubiquitous in the different growth stages of rotifers [30]. In parity with our findings, a recent study also found that Proteobacteria dominate the rotifer-associated bacterial community [21]. Host-related bacterial communities are usually obtained through the horizontal transmission of bacteria existing in the environment [15], but the host is also selective to different bacterial communities [31]. In the present study, Proteobacteria was extremely abundant among *B. calyciflorus*-associated bacteria, indicating that it may be more dependent on the habitat environment provided by *B. calyciflorus*. *Raoultella* belongs to the Enterobacterales, and commonly occurs in the natural environment, such as water, soil, and on plants [32]. The members of *Raoultella* can use glucose as carbon for the co-metabolic degradation of refractory pollutants [33], which has been applied to boost biological performance for the removal of refractory pollutants [34]. In addition, it has been confirmed that it has a fixed nitrogen (N) ability [35]. *Delftia* belongs to the order Burkholderiales, and most of its members have biodegradation effects, such as degrading peptidoglycan and aniline [36,37]. *Delftia* was widespread in rhizosphere soil, activated sludge, and polluted environment [38], and has been reported to be found in the gut flora of fruit fly (*Bactrocera tau*) and white shrimp (*Litopenaeus vannamei*) [39,40]. In this study, the relative abundance of *Raoultella* and *Delftia* in *B. calyciflorus*-associated bacteria was extremely high but extremely low in the natural environmental water, indicating that they relied on the habitat environment provided by the *B. calyciflorus* body, with host specificity. Rickettsiales belong to Proteobacteria and are specialized intracellular parasites that can infect almost all species of a major eukaryotic lineage [41]. *Candidatus Megaira* of Rickettsiales had a high relative abundance in the *B. calyciflorus*-associated bacteria community and was usually associated with ciliate protozoa and has been reported many times in ciliates [42,43]. Although the host was the only resource provider of specific intracellular parasitic bacteria, and parasitic behavior always produces costs, some studies have shown that members of *Candidatus Megaira* can provide positive effects and improve competitive advantage for the host [44]. Furthermore, some members of Pseudomonadales, Rhizobiales, and Gammaproteobacteria incertae sedis mainly involved biodegradation processes, such as phenanthrene [45] and polycyclic aromatic hydrocarbons [46], as well as carbon and nitrogen cycles, such as nitrate reduction [47] and nitrogen fixation [48,49]. All of the above may account for the high relative abundance of Proteobacteria in *B. calyciflorus*-associated bacteria.

Actinobacteriota was the second dominant taxa among *B. calyciflorus*-associated bacterial communities and also had the highest abundance among bacterioplankton in natural water, which was mainly composed of Microtrichales and Frankiales. Its members are highly correlated with important global nitrogen cycle pathways with denitrification and nitrogen fixation [50], such as the *CL500-29 marine group* and the *HgcI clade* [51]. Cyanobacteria had the second highest abundance among bacterioplankton, and it was also the third dominant taxa with *B. calyciflorus*-associated bacteria. We did not empty the intestinal tract of *B. calyciflorus* before the detection of the bacterial community in this study, so the detected associated bacterial community included not only the resident bacteria living in the intestinal tract and the outer surface but also the bacterial community staying briefly in the intestinal tract [12]. The abundant presence of Cyanobacteria in the associated bacteria may result from the inevitable predation of Cyanobacteria by *B. calyciflorus*, especially when cyanobacteria are abundant in the native environment [52]. Verrucomicrobiota, Bacteroidota, and Firmicutes are common bacteria in freshwater [53,54]. The relative abundance of these three taxa in the NB community was higher than that in the AB community, which indicated that these taxa might come from the bacterioplankton community but had a lower level in the AB community due to host-specific selection pressure [55]. Environmental pollutants, such as microplastics and nanoplastics, have been reported to induce bacterial dysbiosis in the guts of *Danio rerio*, with a significant decrease in the relative abundance of Verrucomicrobiota [56]. Both Firmicutes and Bacteroidota have functions in cellulose degradation, carbohydrate, and amino acid metabolism [57,58]. Studies have shown that the ratio of Firmicutes/Bacteroidota was often used as an index of obesity, which was directly proportional to obesity [59], and was related to diet as well [60]. These studies suggested that Firmicutes and Bacteroidota may affect metabolic processes, such as energy, sugar, and lipids, in organisms.

The co-occurrence network is a new means that is often used to speculate and identify the keystone taxa of microorganisms [61,62]. In this study, the network was also selected to speculate on the keystone taxa of bacteria. The results showed that the keystone taxa of *B. calyciflorus*-associated bacteria were composed of Frankiales, Microtrichales, and Gammaproteobacteria_Incertae_Sedis, and the keystone taxa of bacterioplankton were composed of Frankiales, Microtrichales, and Burkholderiales. Numerous members of these three orders have been consistently identified as keystone taxa in different studies and ecosystems [63]. The keystone taxa may play an important role in some processes, such as nutrient cycling and energy flow, thus influencing the ecological functions of bacterial communities that are not absolutely linked to their abundance [64]. Interestingly, the OTUs with a relative abundance of less than 1% were 54.66% in the unique OTUs of *B. calyciflorus*-associated bacteria and as high as 81.64% in bacterioplankton, so we might underestimate the importance of rare taxa in bacterial communities.

### 4.2. Environmental Regulation of AB and NB Communities

Until now, there have been few studies on the differences between AB and NB communities. Studies have shown that the alpha diversity of the AB group was significantly lower than that of the NB communities [12,65]. The community diversity of the AB group was significantly lower than that of bacterioplankton, and their composition was significantly different, so the different microhabitats provided by zooplankton and water were perhaps the main factors causing the difference. PCoA analysis showed that the AB and NB communities were clustered separately, indicating that the bacterial communities in similar habitats were more similar. It has been shown that zooplankton-associated bacteria are not only actively exchanged with habitat bacteria communities, but also that most of them come from the surrounding water environment [11,66]. However, only 40.12% of the same OTUs were found between the two bacterial communities in this study, indicating that the composition of the *B. calyciflorus*-associated bacterial community is very flexible and highly correlated with microhabitat [67]. RDA analysis showed that T, DO, pH, TN, TP, PO_4_^3−^, NH_4_^+^ and NO_3_^−^ were significantly correlated with the variation of AB community, but they on the first two axes explained only 39.55% of the changes with bacterial community. These also indicate that although environmental factors outside microhabitats participate in the regulation of the bacterial community structure, they are not the key factors driving the variation of the *B. calyciflorus*-associated bacterial community. In addition, the diversity, as well as the relative abundance of both bacterial communities, had seasonal variations, and T and NO_3_^−^ may be the main influencing factors [68,69].

The gut of zooplankton can provide good anaerobic conditions for Firmicutes [18]. Members of the Bacteroidota can break down chitin and the chitin exoskeleton of zooplankton by using it as a source of carbon and nitrogen [70]. Free-living bacteria are often exposed to environmental hazards, such as predation, virus cleavage, harmful radiation, and chemicals. The exterior and interior of zooplankton provide specific microhabitats for bacterioplankton, which can provide bacteria with resistance to these external hazards [6]. To sum up, the special microhabitat with selectivity to bacteria provided by *B. calyciflorus* may not only be the main force driving the difference between associated bacterial and bacterioplankton communities but also the main reason why the alpha diversity of associated bacteria was lower than that of the bacterioplankton community.

### 4.3. Main Functions of the B. Calyciflorus-Associated Bacterial Community

The functions of microbial communities, predicted through software such as FAPROTAX, are widely used in microbial ecology [71]. The most important biogeochemical cycle functions of AB and NB communities include plastic degradation, phototrophic, photosynthetic autotrophic, and oxygen-producing photoautotrophic functions, while the most prominent biogeochemical cycle functions of *B. calyciflorus*-associated bacterial community were plastic degradation, chemoheterotrophy, and aerobic chemoheterotrophy.

Plastic pollution has become a global environmental problem, and the degradation of plastics can be divided into abiotic degradation and biodegradation. Many abiotic factors, such as ultraviolet radiation, oxygen, and temperature, are related to the incomplete and long-term degradation of plastics, while bacteria are related to the complete degradation of plastics [72]. Large plastics (>20 mm), microplastics (<5 mm) and nanoplastics (<100 nm) have been proven to cause a series of toxicity to aquatic organisms after entering aquatic ecosystems, but bacteria can use them as carbon sources for growth [56,73]. Actinobacteriota is mainly responsible for degradation under aerobic conditions, while members from Proteobacteria and Bacteroidota are involved in degradation under anaerobic conditions [74]. Studies have shown that microplastics will be ingested by zooplankton or adsorbed on algae for re-feeding and will also adhere to the exoskeleton and appendages of zooplankton, thus having a negative impact on the function and health of zooplankton [75,76]. However, Canniff and Hoang [77] showed that the survival and reproduction of *D. magna* were not significantly affected after ingesting a large quantity of miniature polyethylene beads. Therefore, it is speculated that although the impact of microplastics on *B. calyciflorus* may be deepened due to bioaccumulation [78], the associated bacterial community has improved the tolerance of *B. calyciflorus* to plastic pollution through a specific composition. Chemoheterotrophy and aerobic chemoheterotrophy can use organic matter to meet all or major carbon requirements under different oxygen conditions [79], which are the most dominant putative functions predicted by various biologically associated bacterial communities [71,80,81]. Nitrogen is the other most important element in a lake ecosystem. The increase in nitrogen promotes the growth of phytoplankton, such as Cyanobacteria, which will mass reproduce in summer [82], leading to the deterioration of water sources. Nitrogen excretion by zooplankton can stimulate the growth of bacteria, especially denitrifying bacteria, such as related members of Actinobacteriota and Proteobacteria [10,83]. In the *B. calyciflorus*-associated bacterial community, Proteobacteria and Actinobacteriota were the most abundant taxa, which was consistent with the results of the above research. The abundance of these bacteria in the associated bacteria indicates the importance of *B. calyciflorus*-associated bacteria, owing to their potentially highly significant role in the nitrogen biogeochemistry of lake ecosystems. In addition, human pathogens were abundant in the bacterioplankton community. Environmental water can become contaminated by a variety of sources, such as rainfall and human activities [84,85], so human pathogens can be found in multiple bodies of water [86]. The public health impact of the transmission of human pathogens in water is significant worldwide; hence, it is necessary to explore further in the future.

## 5. Conclusions

Our study illustrated that *B. calyciflorus*-associated bacterial and bacterioplankton communities have different characteristics, and *B. calyciflours*-associated bacteria exhibited high host dependence. *Raoultella* and *Delftia* in Proteobacteria were the dominant groups in the *B. calyciflorus*-associated bacterial community, and they have a good biodegradation function. The functional prediction showed that plastic degradation, chemoheterotrophy, and aerobic chemoheterotrophy were the most prominent functions of the *B. calyciflorus*-associated bacterial community. Moreover, several keystone bacterial members participating in carbon and nitrogen cycles were found in the associated bacterial community by co-occurrence networks, such as Microtrichales and Frankiales. All these results indicate that *B. calyciflorus*-associated bacteria play an important role in host and even freshwater ecosystems. In the future, it is necessary to more intensively study the contribution of zooplankton-associated bacteria to special global biogeochemical cycles, such as pollutant degradation.

## Figures and Tables

**Figure 1 animals-12-03201-f001:**
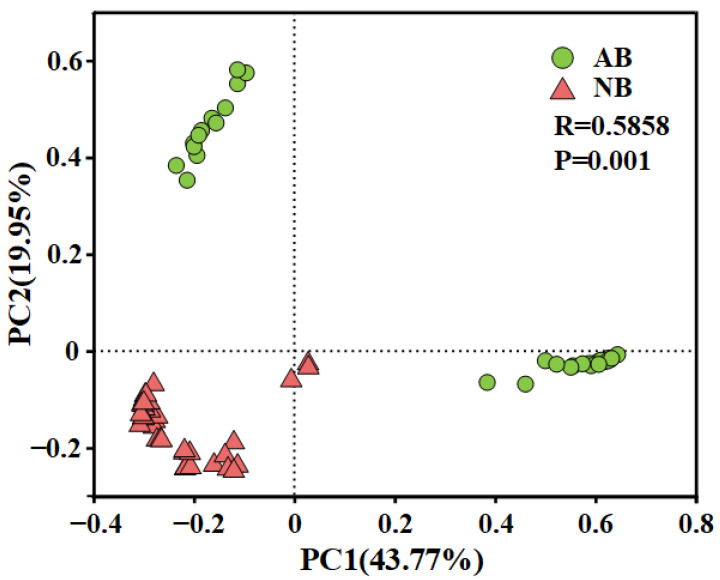
PCoA principal coordinate analysis of *B. calyciflorus*-associated bacterial and bacterioplankton communities.

**Figure 2 animals-12-03201-f002:**
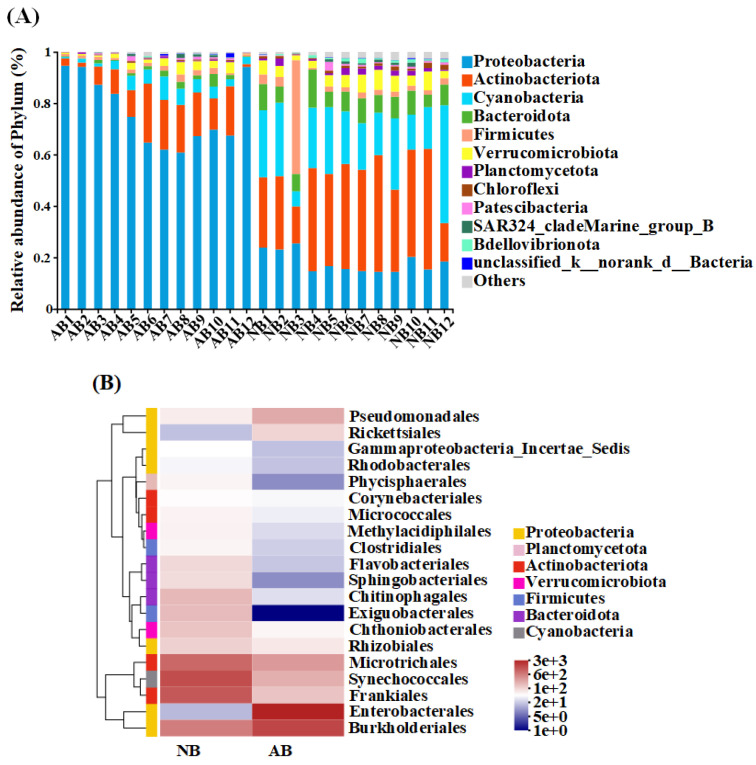
Relative abundance of the community composition of *B. calyciflorus*-associated bacteria and bacterioplankton. (**A**) Community barplot analysis at the phylum level. (**B**) Community heatmap analysis on the order level.

**Figure 3 animals-12-03201-f003:**
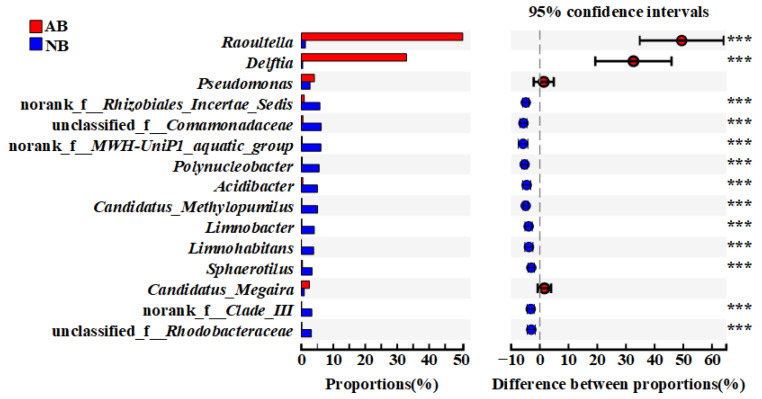
Differences in the top 15 genera in Proteobacteria among *B. calyciflorus*-associated bacterial and bacterioplankton communities. The analysis was based on the Student’s *t*-test. *** *p* ≤ 0.001.

**Figure 4 animals-12-03201-f004:**
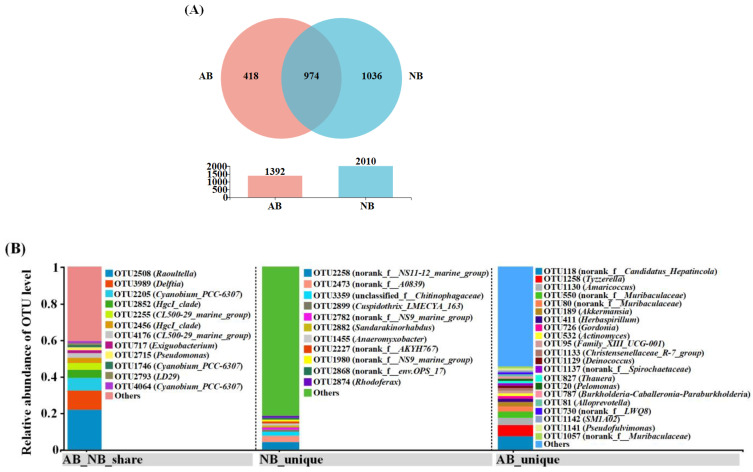
Distribution and quantity of unique or shared OTUs in *B. calyciflorus*-associated bacterial and bacterioplankton communities. (**A**) The number of unique or shared OTUs. (**B**) Distribution of unique and shared OTUs.

**Figure 5 animals-12-03201-f005:**
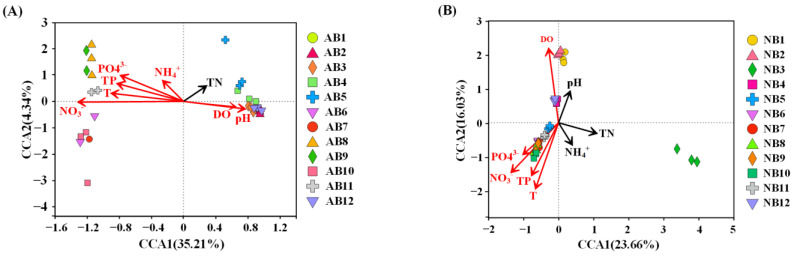
The canonical correspondence analysis (CCA) plot investigates correlations between bacterial communities and environmental factors at the OTU level. (**A**) *B. calyciflorus*-associated bacteria community; (**B**) bacterioplankton community. Each significant factor (*p* < 0.05) is shown in the plot with red arrows.

**Figure 6 animals-12-03201-f006:**
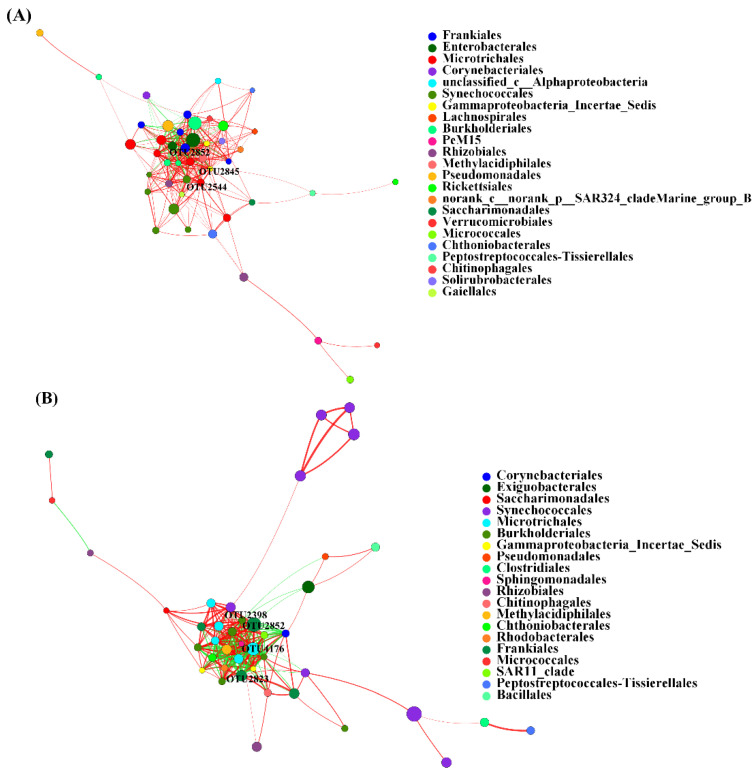
The OTU co-occurrence network of *B. calyciflorus*-associated bacterial and bacterioplankton communities. (**A**) *B. calyciflorus*-associated bacteria (AB); (**B**) bacterioplankton (NB). The nodes are colored according to order. Green edges represent positive correlations, and red edges represent negative correlations. Node size is proportional to the betweenness centrality of each OTU, and edge thickness is proportional to the weight of each correlation.

**Figure 7 animals-12-03201-f007:**
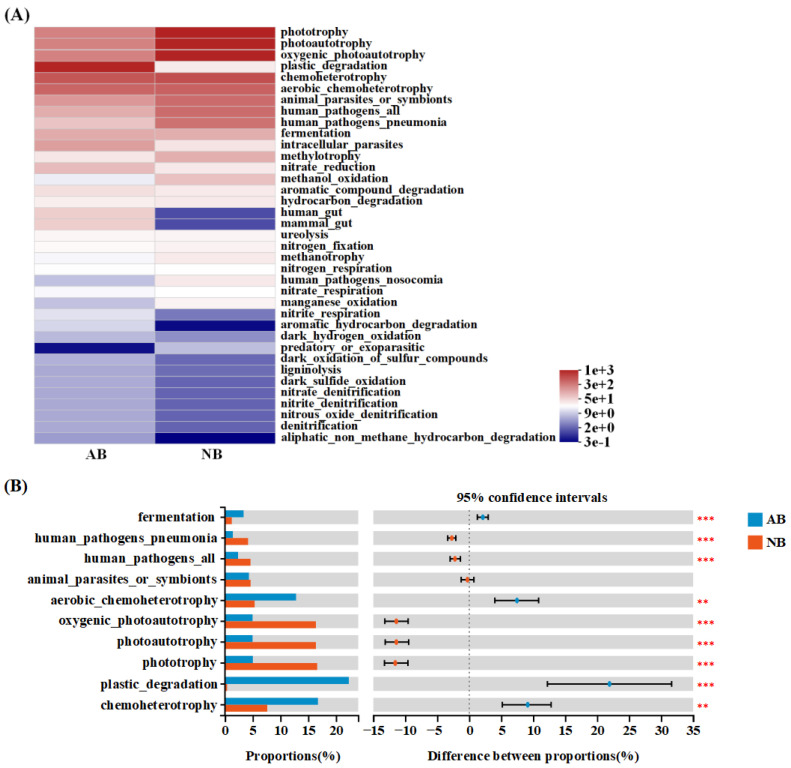
Functional distribution of *B. calyciflorus*-associated bacterial and bacterioplankton communities based on FAPROTAX function prediction. (**A**) FAPROTAX function consequential heatmap; (**B**) FAPROTAX functional groups difference test. Showing the significant difference in functional groups of abundance in the top 10. The left shows the abundance ratio of different functional groups; the middle shows the percentage of functional group abundance differences within a 95% confidence interval. The right, ** *p* < 0.01 and *** *p* < 0.001.

**Table 1 animals-12-03201-t001:** Alpha diversity index of *B. calyciflorus*-associated bacteria (AB) and bacterioplankton (NB) (Wilcoxon rank-sum test).

Estimators	AB (Mean ± SD)	NB (Mean ± SD)	*p* Value
Shannon	1.86 ± 0.83	4.27 ± 0.41	<0.001
Chao	237.26 ± 114.92	769.47 ± 188.45	<0.001
Coverage	0.99 ± 0.01	0.97 ± 0.01	<0.001
Pd	26.48 ± 9.02	57.69 ± 11.80	<0.001

## Data Availability

Data are available from the China National Center for Bioinformation (GSA number: CRA008052) that are publicly accessible at https://ngdc.cncb.ac.cn/gsa (accessed on 6 July 2022). Any further data required can be requested from the corresponding author.

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
