# Peer review of "The Relationship between Brachionus calyciflorus-Associated Bacterial and Bacterioplankton Communities in a Subtropical Freshwater Lake"

_animals, 2022, doi:10.3390/ani12223201_

Round 1

Reviewer 1 Report

The authors have done identifying relationships between Brachionus calyciflorus-associated bacterial and natural bacterioplankton communities structures, their predicted functionality, and analyzed their co-occurrence networks in subtropical freshwater lake and determining environmental drivers that influence the structure of the communities.

I have some questions about sampling, environmental parameters, high throughput sequencing data analysis and statistical approach of correlation-based network.

 Abstract

Line 57. “has similar but different community composition” Could you use a clearer phrase?

Introduction

Line 58. [8;9] should be changed to [8,9]

Line 96-98. Reference [23] is located after the phrase “… transfer of energy in subtropical freshwater lake [23]”, but [23] - Jensen, et al. 2004 does not contain anything about a subtropical freshwater lake. This reference should probably come after another part of the sentence “B. calyciflorus was used to be the model organism in this study”. Because Jensen, et al. 2004 laboratory cultivation of B. calyciflorus studied. A little unclear, please write more clearly.

 2. Materials and Methods

Line 105-106. What is the total depth at the sampling station?

Line 107-111. You have determined every month water temperature (T), dissolved oxygen (DO) and pH, TN, TP, PO43−, NH4+ and NO3−. In chapter 3.4 you describe how communities relate to these parameters. But in the results there is no description of quantitative data for these parameters anywhere. How did the temperature change during the year? What physical and chemical conditions were in the study period? A table with sampling dates and physico-chemical parameters for these samples can be placed in supplementary materials.  

Line 117-126. Could you describe the sampling in more detail. From what depths were zooplankton and water samples taken? How were water samples taken?

Line 124. You recruited 500 rotifers for rotifer-associated bacteria analysis. Did you count the number of rotifers in the samples? Does the number of rotifers change during the year?

Line 130. For the study of B. calyciflorus-associated bacteria, did you isolate total DNA from a collected biomass of B. calyciflorus (500 similar individuals in each sample)? Have you isolated DNA from bacteria that are outside and inside (in the intestinal tract) of rotifers? This should be explained.

Line 146. Usually, singletons (OTUs represented by a single sequence) are also removed from such data, which are most often sequencing errors. After removing singletons, the number of OTUs in your data can be significantly reduced.

Line 149. Chloroplast and Mitochondria need to be removed from the data of 16S rRNA as they are not bacteria but parts of eukaryotic phytoplankton.

Line 154-155. Why do you additionally use Qiime to calculate beta diversity distance matrix, if Mothur, in addition to calculating alpha diversity index, can also calculate beta diversity? You are using the RDP classifier for classification of each OTU representative sequence. But Mothur and Qiime create OTUs and also use RDP classifier for classification of OTU representative sequence. Therefore, it is not clear why use different programs if you can perform all the functions using one of them.

Line 157-158. In various articles using correlation analysis and networking, the term “co-occurrence network” or “correlation network” is most often used, rather than “co-correlation network”.

Line 158. Could you add a link to FAPROTAX.

Line 162. “pearson” should be changed to “Pearson”

Line 162-163. “analysis of multiple comparisons and analysis of pearson correlation were used to analyze the diversity index and environmental factors of zooplankton.” Chapter 3. Results does not contain information on the results of the Pearson correlation of diversity index and environmental factors analysis.

3. Results

Line 174. “were obtained from 72 samples” - what are these 72 samples? How many samples did you take in total for analyzing bacterial communities? If you sampled every month for a year, then 12 samples were taken for AB and NB. This corresponds to Figure 3A, which has 12 samples for AB and 12 samples for NB. There are 24 samples in total. Clear it up.

Line 186-191. Table 1 shows the average alpha diversity indices for all samples AB and ND? Were there any seasonal differences in diversity among AB and NB communities? Information on alpha diversity for all samples for AB and NB communities is best placed in Supplementary material, since it is important to know how diversity changed during the year.

Line 192. “Figure 1. Sparse curve…” would be better moved to Supplementary material as this information is of secondary importance for the biological understanding of bacterial communities.

Line 196-198. The phrases “and the first coordinate accounts for 43.58% of the sample variance, and the second coordinate accounts for 20.26%” should be deleted because this information is shown in Figure 2.

Line 205. “Proteobacteria (77.98%)” Does this share represent the ratio of reads for all samples?

Line 215. Chloroplasts need to be removed from the data. Chloroplasts are not bacteria, but parts of eukaryotic phytoplankton cells.

Line 219. Figure 3 A. In the diagram, Actinobacteriota and Planctomycetota have almost the same red color. It is better that they differ in more contrast.

Line 233. “After processing the data, the total 3502 OTUs were obtained.” - this information is already indicated on Line 179. Do not duplicate information.

Line 238-240. “Raoultella OTUs (OTU2508), Delftia OTUs (OTU3989), Cyanobium PCC-6307 OTUs (OTU2205, 2255, and 4176), and hgcI clade OTUs (OTU2852 and 2456)” should be changed to ““Raoultella (OTU2508), Delftia (OTU3989), Cyanobium PCC-6307 (OTU2205, 2255, and 4176), and hgcI clade (OTU2852 and 2456)” because in this style you have written the following text.

Line 246. Chloroplast need to be removed.

Line 249. You analyzed the composition of bacterial communities every month for a year (12 time points). However, nothing is said about seasonal changes in the composition of communities. Does the composition of communities remain the same in all seasons? Are there absolutely stable physical and chemical conditions in this lake?

Line 250. In figure 5 (B, C, D) it would be better if the name of the taxon was written next to the OTU numbers. OTU numbers are not convenient to read without taxa. It is possible that in order for the names of taxa to fit compactly in the figure, the pie charts should be replaced with charts of a different type.

Line 257. How is VIF decrypted?

Line 269. You analyzed the relationship between physicochemical parameters and community structure. However, the article lacks quantitative data on these parameters. Therefore, it is not clear what this relationship is.

Line 270. In Figure 6 there is no temperature. Temperature was not a significant factor in changing the structure of communities?

Line 276-277. You have computed Spearman's correlations between OTUs. But there is nothing in the methods about how this correlation analysis was carried out. Correlations with what coefficients did you use to build the network and what correlation correction did you perform? To perform good correlation analysis, you need to remove the compositionality property of the data (because you are using relative abundance) by using a logarithm. After calculating Spearman’s correlations, it is necessary for the correlations to have P values ​​of at least 0.05. P values ​​for the correlation coefficients should be corrected for false discovery rate in multiple comparisons using, for example, the Benjamini–Hochberg equation or other methods of corrections. After removing compositionality and adjusting p values, the number of correlations is significantly reduced, but this will be a good accurate result. After performing such an analysis, the number of correlations will most likely be small, given that you have a small sample (only 12 samples). Therefore, the correlation network will change and all network parameters will change.

Line 293. The networks contain Chloroplasts. They must be removed. If correlations were made on the basis of relative abundance of OTUs, then after removal of chloroplasts, correlations for other OTUs may change. Is it possible to somehow improve the quality of edges in the network and make solid lines out of dotted lines.

Line 313-314, 316. Why among the functional groups are Cyanobacteria and chloroplasts, which are not a function.

4. Discussion

Line 347-348. Phrase and link mismatch: “White shrimp and Drosophila melanogaster [35,36].” 35 – Prabhakar et al., 2013 (fruit fly Bactrocera tau); 36 – Wu et al., 2021 (white shrimp Litopenaeus vannamei).

Line 351. Rickettsiales also includes mitochondria. Therefore, if you did not remove the mitochondria, then the Rickettsiales ratio are most likely distorted (Fig. 3 B, Fig. 7 A). In my study, I recently analyzed freshwater bacterial communities and found that all 29 Rickettsiales OTUs (silva taxonomy) in my dataset turned out to be Mitochondria, which of course need to be removed.

Line 387. “Co-correlation network” should be changed to “co-occurrence network” or “correlation network”.

Data Availability Statement:

Line 491-493. I went to https://ngdc.cncb.ac.cn/gsa, then I typed GSA number: CRA008052 and I got the following entry: “GSA: CRA008052 Accession: CRA008052 Title: CRA008052 BasicInfo : PRJCA011475; CRA008052; The data under accession CRA008052 will be available on 2023-09-01.” There is no more information. There are no tabs to view raw sequence data. Therefore, the sequence dataset from this article is not available for free viewing. Please clarify this situation.

Dear authors, let me remind you that the removal of singletons, chloroplasts and mitochondria will lead to a change in the diversity indices and the ratio of the shares of other taxa, which will lead to a change in the statistical results, so many figures and text will need to be changed. Perhaps 7 days for revisions of this manuscript is too short to carry out such a large amount of work. So if you need more time, I think you should tell the editor to carefully correct and check everything.

Reviewer 2 Report

The authors present a descriptive work on the bacteria associated with the rotifer Brachionus calyciflorus in comparison with the surrounding bacterioplankton. I have several main concerns that I present here below.

My first consideration is minor but requires adapting all over the text the term “natural bacterioplankton”. It seems the authors mean to differentiate free-living bacterioplankton in comparison to the organisms that are found in association with the rotifer. If this is the case, there are two options: either calling in free-living or simply stating bacterioplankton. No need for the “natural” adjective. On the other hand, if the authors refer to the native organisms in the geographical location, the words “native” or “endemic” should be better.

Secondly, and even though the work presents good methodology, there is a need for major English revision. There are too many errors of subject-verb agreement and expressions that need to be rewritten.

The soundness of choosing B. calyciflorus was very briefly described in the introduction. Could the authors present numbers regarding the proportion of prevalence of this rotifer in their samples and overall? In the same last paragraph of the introduction, it is missing a hypothesis for this work. What is expected to be seen regarding the effect of the environmental factors on the bacterioplankton associated or not to this model rotifer? Do we have already information on the effect of the environment on the rotifer itself?

In terms of the methodology of sampling, it is not clear how the authors are sure the have sampled only this species and not a whole community of rotifers. If any taxonomic macroscopic characteristic was used to differentiate and pick the individual organisms, please explain. There is also a lack of information on the source of the associated bacteria. Would they be colonizing the rotifer surface or can they also be from the rotifer gut microbiome ?

The methodology is in accordance with the objectives, and it is well explained for the sequencing analysis. However, I wonder if any filter was used before analysis, for example to filter out archaea or singletons (rare OTUs). It also came to my attention that the authors have used mothur for alpha-diversity, and qiime for beta-diversity. Please clarify if there was a specific reason for doing so.

molecular analysis lacks some important information as how many soil samples were used for sequencing, and how many reads were obtained. Also, the description of sequencing results could benefit of the use of richness concepts and calculations, instead of just basing the discussion on the raw numbers of OTUs.

For example, we can see that chloroplast was kept unfiltered and the authors refer to this erroneously in the same manner as other taxonomic order. The classification of chloroplast refers to the presence of plants (algae) in the sample. Most studies describing a microbiome exclude chloroplast OTUs from the analysis. The authors can either do this filtering or explain the reasoning to keep this sequences and exploit them separately from other bacterial groups.

Furthermore, be careful about sentences of conclusion that are beyond the obtained results. For example, in the abstract, there is no indication that “associated bacteria can supplement the deficiency of these biogeochemical processes in the ambient waters.”

I present here a non-exhaustive list of minor considerations. Please correct all species name that are not in italic as for example in lines 72 and 73, but also in table and figures legends.

In line 89, do the authors mean “culture stability and population growth” in laboratory? Could this be specified?

In lines 198 to 200 the authors use the exact same sentence as in the abstract. Please modify one or the other.

Line 235 indicate there is a difference between the associated and non-associated communities which is highly expected, almost obvious. Not sure there is need to state this.

Figure 1 represents the rarefaction curve which is very rarely used still. This could go to the supplementary material or even completely excluded since we have the coverage numbers in the previous paragraph.

Figure 5 should be present otherwise. Another heatmap or barchart could be more intuitive to see all (B) (C) and (D) results. Instead of seen the OTUs names it would be also more interesting to have taxonomic information.

Line 314. Do the authors have an explanation for the presence of human pathogens bacteria in the NB portion? Was this expected from the type of sample? This sounds as contamination from human sample manipulation. Is this a possibility?

The discussion regarding the source of associated bacteria can be added when the authors present the known niches from the bacteria that were detected in higher abundance in the AB compartment. Lines 337 to 361.

Once more, be careful about extrapolations or conclusions that are neither supported by the data or by the literature. Could you provide more information on line 368 that says that the rotifer was fed on cyanobacteria?

Round 2

Reviewer 1 Report

The authors of this article have done a good job of improving it. There are a few minor remarks.

 1. Introduction

Line 101. Phrase “and more” should be deleted.

 3. Results

Line 203-204. “…457 family, 840 genus, 1379 species”. I think most of the 1379 species obtained are unclassified. Because the database used contains a small number of sequences identified to the species. Therefore, the phrase “1379 species” should be removed.

 Table S1. – The name of the table is “AB and NB communities…” But the names AM and NM are used in the table. Table S2 has the same names. What does AM and NM mean? There is no decryption anywhere.

 Line 230. “bray-curtis” should be changed to “Bray-Curtis”.

Line 238-239. “Actinobacteria” – you probably meant the phylum Actinobacteriota. Correct the title.

Line 289. “(Figure 4BB)” should probably be corrected “(Figure 4B)”

Line 334. Could you added the total number of correlations for AB and NB. How many of them had negative and positive correlations. This is important information when comparing networks.

Reviewer 2 Report

I thank the authors for the corrections made. 

Please find below some more minor points to be consider before going ahead.

Line 13. zooplancton HAS an important role...

Line 15. ammonia oxidation is part of the nitrogen cycle. Consider deleting the "ammonia oxidation".

Even though the manuscript is improved by having the presented hypotheses, these can still be changed for greater precision. The first hypothesis is on the past which gives the reader an idea that it was written after the results were obtained. Change the verb tense to the infinite as the other two hypotheses.  They are still hypotheses which are quite evident and lack a bit more of detail on the "because" of what should be found. Having said this, one must acknowledge that it is difficult to set hypothesis for descriptive studies as this one. 

All modifications brought to the methods make it much more easier to understand the details of the analyses. The discussion has been greatly improved as well. 

Line 231. Delete "obviously". Change to "taxonomically distant", instead of far away
